# A Cluster Approach to Cloud Cover Classification over South America and Adjacent Oceans Using a k-means/k-means++ Unsupervised Algorithm on GOES IR Imagery

**Adrián E. Yuchechen** [1,*] **, S. Gabriela Lakkis** [2] **, Agustín Caferri** [3] **, Pablo O. Canziani** [1] **and Juan Pablo Muszkats** [4,5]

1   Unidad de Investigación y Desarrollo de las Ingenierías (UIDI), Consejo Nacional de Investigaciones Científicas y Técnicas (CONICET), Facultad Regional Buenos Aires (FRBA), Universidad Tecnológica Nacional (UTN), Ciudad Autónoma de Buenos Aires (CABA) C1407IVT, Argentina; pocanziani@frba.utn.edu.ar
2   Facultad de Ingeniería y Ciencias Agrarias, Pontificia Universidad Católica Argentina, UTN, FRBA, UIDI, CABA C1107AFD, Argentina; gabylakkis@uca.edu.ar
3   UTN, FRBA, UIDI, CABA C1407IVT, Argentina; acaferri@frba.utn.edu.ar
4   Departamento de Matemática, Facultad de Ingeniería, Universidad de Buenos Aires, CABA C1063ACV, Argentina; jpmuszkats@comunidad.unnoba.edu.ar
5   Departamento de Ciencias Básicas y Experimentales, Universidad Nacional del Noroeste de la Provincia de Buenos Aires, Junín B6000FJJ, Buenos Aires, Argentina
*   Correspondence: aeyuchechen@frba.utn.edu.ar; Tel.: +54-11-4867-7268

**Abstract:** An unsupervised k-means/k-means++ clustering algorithm was implemented on daily images of standardized anomalies of brightness temperature ($T_b$) derived from the Geostationary Operational Environmental Satellite (GOES)-13 infrared data for the period 1 December 2010 to 30 November 2016. The goal was to decompose each individual $T_b$ image into four clusters that captures the characteristics of different cloud regimes. The extracted clusters were ordered by their mean value in an ascending fashion so that the lower the cluster order, the higher the clouds they represent. A linear regression between temperature and height with temperature used as the predictor was conducted to estimate cloud top heights (CTHs) from the $T_b$ values. The analysis of the results was performed in two different ways: sample dates and seasonal features. Cluster 1 is the less dominant one, representing clouds with the highest tops and variabilities. Cluster 4 is the most dominant one and represents a cloud regime that spans the lowest 2 km of the troposphere. Clusters 2 and 3 are entangled in the sense that both have their CTHs spanning the middle troposphere. Correlations between the monthly time series of the number of pixels in each cluster and of the entropy with several circulation indices are also introduced. Additionally, a fractal-related analysis was carried out on cluster 1 in order to resolve cirrus and cumulonimbus.

**Keywords:** brightness temperature; cloud cover; cloud regimes; clustering; GOES IR imagery; kmeans; kmeans++; South America

## 1. Introduction

In the recent years the scientific community has focused on cloud cover (CC) because of its role in the Earth's radiative balance (ERB). In 2013, the Intergovernmental Panel on Climate Change (IPCC) included for the first time a dedicated chapter to this topic alongside aerosols [1]. Clouds critically influence the radiation budget since they reflect a substantial portion of the incoming solar radiation

back to space, and partly trap the outgoing terrestrial radiation and re-emit it to space at the temperature of their tops [2]. Even though this influence can be theoretically determined by CC's density, height and thickness (among other parameters) the result can be disappointing since clouds scatter, absorb and reflect radiation in numerous complex ways. Furthermore, not only the presence of clouds but also their formation/evolution contributes to the ERB since the release of latent heat owing to condensation during the formation process heats the atmosphere [3]. Consequently, a number of research efforts have been devoted to address the connections between CC and climate and climate change [2,4–10].

Time series of CC data from conventional information collected by meteorological observers all around the world are prone to subjective errors [11] and cannot be automatically performed. Subjective observations supplemented by data from ground equipment measurements using various technologies that can be implemented in an automated fashion allows for more detailed studies, but they are usually limited spatially, temporally or both [12,13]. Identifying the presence and behavior of distinct clouds is a manifold task that requires the understanding of the different meteorological systems and their associated phenomena, and also involves the development of products and techniques that can help assessing their spatial and temporal behavior using remotely sensed data [14–18].

Object detection is the main application for satellite imagery (SI) ([19] and references therein). In particular, in the last four decades it became one of the most powerful and useful sources for obtaining meteorological information relevant to forecasting and to analyze different weather events [20–25]. Clouds generally have both a higher reflectance and a lower temperature when compared with the Earth's surface and this distinction facilitates their detection by satellite sensors. SI has been used to establish cloud microphysical [26] and macrophysical properties [27] and cloud-base heights and temperatures [28,29], among many other applications.

The Geostationary Operational Environmental Satellite (GOES) is a geostationary satellite network consisting of a two-spacecraft constellation operated by the National Oceanic Atmospheric Administration (NOAA) [30]. At January 2005, GOES-12 and GOES-10 operated as GOES-E and GOES-W, respectively. They were positioned at 75 °W (over North-Western South America) and 105 °W (over the central Pacific), respectively, and observed 60% of the Earth [30]. GOES-13 replaced GOES-12 as GOES-E in April 2010 [31]. The Imager instrument on board the different GOES spacecraft from GOES-12 to GOES-15 sensed the atmosphere in five spectral bands, four of them with their central wavelengths within the infrared (IR) spectrum, and the remaining one in the visible (VIS) spectrum [32] (pp. 11–12). Considering specific CC studies, GOES-2 IR data was used to study its distribution across the Americas ([33] and references therein), GOES-8 and -9 IR data was processed in order to track the motion and evolution of stratiform clouds [34], and GOES-11 imager and sounder information was taken to describe the outbreak of a severe thunderstorm over the United States Great Plains [35]. CC was also analyzed for masking cloud imagery so as to detect fires [36].

The deficient in situ cloud measurements in South America (SA), the lack of records in its surrounding oceans, and the fact that the GOES-E dataset covers both land and oceans spatially in a continuous fashion since the mid-1970s [30,37,38] point to the use of GOES retrievals to study CC evolution in the region. As computer technologies evolved data classification experienced a gradual transition from the traditional, time-consuming statistical modeling and pixel-by-pixel analyses to distinct mathematical methodologies that ingest and process satellite data in a timely manner. Concerning GOES imagery, the main setback of traditional point-by-point analyses, which establish if a pixel is cloud-covered or cloud-free by comparing its value with a prescribed threshold, is that it is far too inefficient in handling a large set of GOES images due to the huge amount of information a single image has (e.g., the number of pixels in each image of this study is above 5 million). There is a wide range of cloud classification algorithms applied to GOES imagery. Support vector machine classifiers [39], neural networks [40] and supervised algorithms ([41] and references therein) are just a few examples of them. Even though these methods can be adapted (possibly with modifications) to any region covered by GOES data, most of them were applied to CC analyses in areas located in the Northern Hemisphere, partly because of the number of validation opportunities the research

community is presented with when compared with the Southern Hemisphere (SH). With a limited number of exceptions (e.g., [42]) no algorithms have hitherto been applied to, or particularly developed for the analysis of cloud systems in SA and adjacent oceans using GOES retrievals. The main reasons for undertaking an automated classification method for meteorological applications is thus linked to the huge volume of data that is available every day from the satellite and to the lack of studies in the region. In certain cases the analysis of several consecutive images is crucial to forecasting and nowcasting (e.g., severe weather events). As pointed out in [43] there is yet another motivation for this paper and it relies upon the identification of synoptic and subsynoptic cloud systems that follow the analysis of individual images. In order to provide further insights on CC classification in general, and for the SH in particular, this study is based upon an unsupervised k-means clustering algorithm with a k-means++ initializing algorithm that is individually applied on daily IR GOES-13 brightness temperature ($T_b$) data over Southern SA and the contiguous oceans for the 2010–2016 time span. The k-means algorithm has been extensively used to classify cloud top pressure/optical thickness joint histograms in order to obtain cloud regimes [44–47]. This k-means classification was extended with the incorporation of IR information [48]. All these studies were applied to gridded data. The current paper paves the way for the analysis of sequential images in the study region on a pixel-by-pixel basis, as well as for the characterization of synoptic and subsynoptic processes from a CC perspective.

## 2. Materials and Methods

### 2.1. Data

The GOES-13 dataset used in this study was obtained from the NOAA's Comprehensive Large Array-Data Stewardship System (CLASS, available at http://www.class.noaa.gov). Spanning the period 1 December 2010 to 30 November 2016 (2192 days), daily NetCDF-formatted channel 4 IR images taken at 11:45 UTC were selected for each of the days of the study period. Each pixel in the selected imagery represents an area of 4 km by 4 km; the central IR wavelength corresponds to 10.70 μm with a bandwidth of 1 μm [32] (pp. 11–12). Availability was over 97%, totaling 2136 images.

The basic IR quantities measured by the satellite sensors are raw digital counts, which are routinely converted to scaled radiances (SRs) packaged in 10-bit words in a format termed GVAR (for GOES Variable Format) [49]. Matlab's ncread function was used to read the GVAR values from the NetCDF files. A transformation of the GVAR values, i.e., calibration, is mandatory so as to express the SRs into a physical quantity. This calibration was carried out following NOAA's procedure to transform SR into $T_b$ [49]. It represents the intensity of the radiation at a certain frequency, and in this paper is the working variable. The calibration process requires three intermediate steps. The first one calculates the scene radiance (R). Expressed in mW m$^{-2}$ sr$^{-1}$ cm, R is calculated from SR using the linear transformation

$$R = \frac{SR - b}{m} \tag{1}$$

The relationship in (1) is extensive to all GOES IR channels; for the particular case of channel 4, the values of the scaling slope m and the scaling intercept b are 5.2285 and 15.6854, respectively [49]. The second calibration step transforms the R values into an effective temperature $T_{eff}$ using the inverse of the Planck function as follows

$$T_{eff} = \frac{c_2 v}{\ln\left[1 + \frac{c_1 v^3}{R}\right]} \tag{2}$$

In (2) v stands for the central wavenumber of the IR channel 4 $\left(v = 937.23449 \text{ cm}^{-1}\right)$ and $c_1$ and $c_2$ are equal to $1.191066 \times 10^{-5}$ mW m$^{-2}$ sr$^{-1}$ cm$^4$ and 1.438833 K cm, respectively [49]. The final transformation is the one from $T_{eff}$ to $T_b$, which is made through a quadratic function

$$T_b = \alpha + \beta T_{eff} + \gamma T_{eff}^2 \tag{3}$$

For the IR channel 4 $\alpha$, $\beta$ and $\gamma$ in (3) are equal to $-0.52227011$, $1.0023802$ and $-2.0798856 \ 10^{-6}$, respectively [49].

It should be noted that the retrieval of some cloud properties from IR measurements at selected channels is sensitive not only to these desired properties but also to a number of atmospheric and surface parameters that may vary both spatially and temporally [50]. As a consequence, the $T_b$ values obtained through the aforementioned set of equations are subject to certain limitations due to the inherent sensitivity of the observed IR radiances. Measurements in the VIS and in the IR spectra can be overwhelmed by thicker/lower water clouds and so thin cirrus clouds have faint signals that are difficult to detect in both channels [51]. Following this, cirrus cloud top temperatures can be biased towards higher values and in consequence their top altitudes can be underestimated [52]. Moreover, subtle differences can emerge from the detection related to cloudy or cloud-free pixels depending on the band [53]. Further discussions regarding the use of a single IR channel are given below.

Each of the original 2136 individual images was transformed into $T_b$ images following the calibration process. Each resulting image was checked for consistency, and some of them were discarded for having either one or both of the following issues: (a) incomplete raw data information related to noise (49 images, 2.29%) and (b) images that have an anomalous scale of $T_b$ values (51 images, 2.39%). Examples of images with these two issues are shown in Figure 1. Excluding the images having issues (a) and (b), which accounted for 4.68% of the total, the number of analyzable images was 2036. Pixels that populate the lower blank portions at both sides of the original $T_b$ images have no valid latitudes or longitudes as they are beyond the edge marked by the Earth's curvature (see, e.g., Figure 1c). These pixels were removed from each original image prior to the analysis, so the input data consisted of non-zero values only.

Prior to the clustering analysis, and in order to homogenize the set of usable images, each one of them was applied a month-by-month and pixel-by-pixel subtraction of the corresponding monthly mean and divided by the corresponding monthly standard deviation (SD). Figure S1 shows the monthly mean values and SDs. The homogenization process makes the input images comparable to each other. In particular, it removed the meridional gradient that is present in the mean values (cf. Figure S1).

*2.2. The k-means/k-means++ Clustering Algorithm*

Clustering is a widespread technique that is used in many applications, such as in data mining ([54] and references therein), in pattern recognition [55,56] and in machine learning problems ([57] and references therein). In particular, the k-means algorithm [58] is an exploratory procedure that is extensively used in pattern recognition and clustering ([59] and references therein). Given a field with data points $p(i,j)$, $i \le I$, $j \le J$ as input and an integer value K known a priori, the k-means method consists in finding K cluster centers (or centroids) so that the points assigned to a centroid are closer to it than they are to any other. The assignment is unique and the centroid values are usually obtained by weight-averaging all the points identified with this centroid [54]. Despite a number of shortcomings, simplicity and speed of convergence towards an optimal solution make this method one of the most useful across all disciplines [54]. Nevertheless, the performance of k-means was improved by combining it with the k-means++ initialization algorithm, which is a subprocess that seeds the centroids [60]. To this particular aim, k-means++ outperforms k-means both by achieving the classification task quicker and by a faster convergence to a minimal intra-class (intra-cluster) variance [60,61].

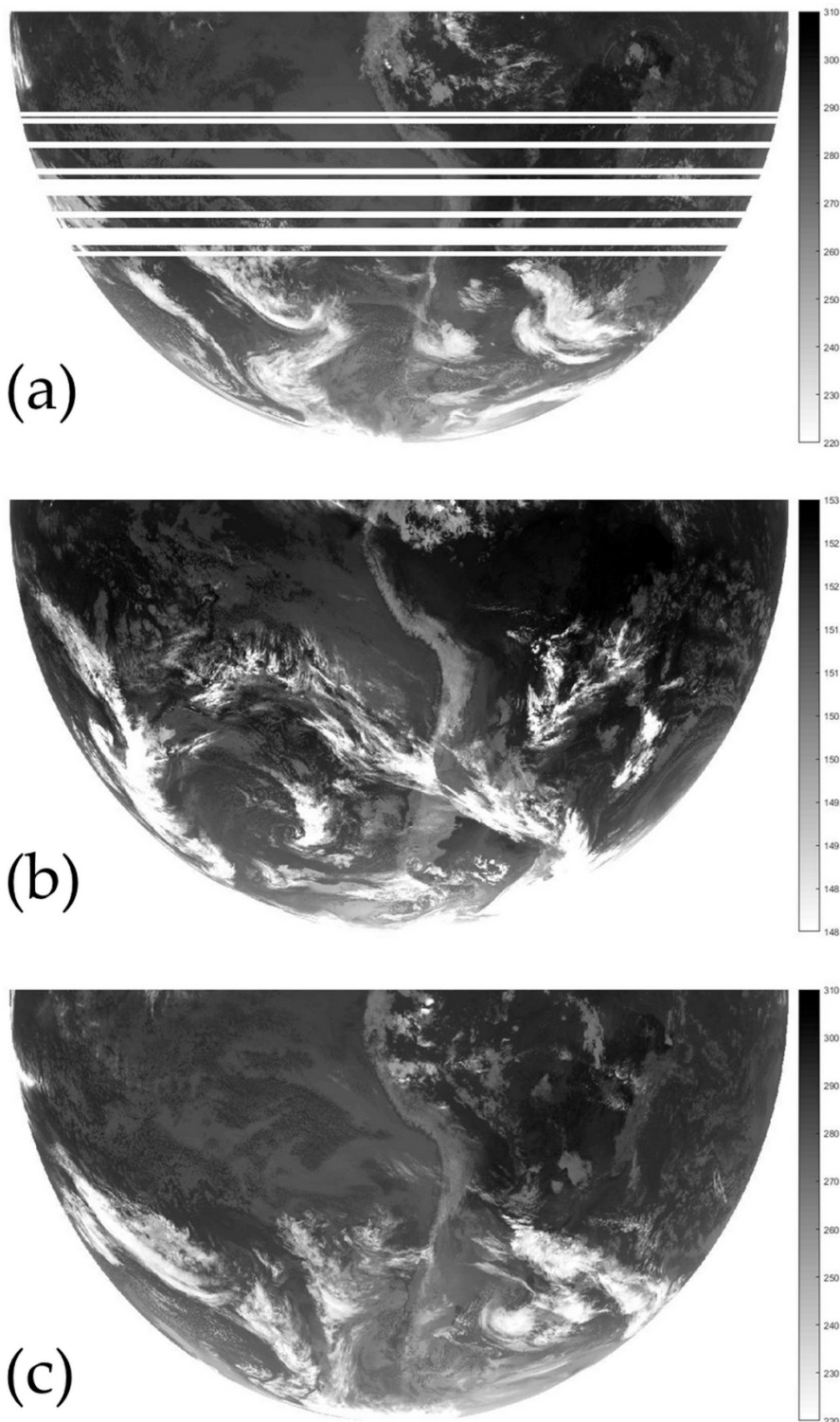

**Figure 1.** Examples of (**a**) an incomplete image with blank rows and (**b**) an image with an anomalous brightness temperature ($T_b$) scale. A regular image is presented in (**c**) for comparison purposes. Unlike (**b**), the image in (**a**) has the $T_b$ scale unaltered when compared with (**c**). Scale values are in Kelvin.

The k-means++ algorithm operates as follows:

1. A pixel from $p(i, j)$ is randomly selected as the first centroid $C_1$;
2. All individual Euclidean distances from each pixel to $C_1$, denoted as $d(p, C_1)$, are computed;
3. The second centroid $C_2$ is randomly selected with probability.

$$\frac{d^2(C_2, C_1)}{\sum_p d^2(p, C_1)} \tag{4}$$

Equation (4) shows that the farther the location of $C_2$ from $C_1$ the higher the probability of selecting it as the second centroid.

4.  The process is repeated until all K centroids are obtained. Similar to step 3), the k-th centroid $C_k$ $(3 \le k \le K)$ is selected from $p(i, j)$ with probability.

$$\frac{\sum_\alpha^{k-1} d^2(C_k, C_\alpha)}{\sum_p \sum_\alpha^{k-1} d^2(p, C_\alpha)} \tag{5}$$

A successful determination of the K centroids carries with the condition of the overall distance between each other being a maximum. Once the K centroids have been determined with the aid of k-means++, the process continues as with the standard k-means algorithm:

5.  The distances $d(p, C_k)$ for $k \le K$ are computed;
6.  Each grid point is assigned to the cluster with the closest centroid;
7.  For $k \le K$, the average distance of all the pixels belonging to the $k - th$ cluster is calculated so as to reassign this value to the corresponding centroid.

The k-means process was repeated until the maximum number of predefined iterations was reached or when cluster assignments no longer changed. The kmeans/kmeans++ technique was individually implemented on each of the 2036 homogenized images using Matlab's kmeans and kmeans++ functions. Several replicates with random starting seeds were generated in order to avoid local minima and overfitting problems.

The establishment of the optimal value for the unknown parameter K that minimizes misclassification is a frequent issue. To this respect, the Caliński–Harabasz criterion (CHC) [62] was used here in order to estimate it. CHC establishes a variance ratio criterion (VRC) from which the optimal number of clusters K is estimated using the following quotient

$$VRC = \frac{BGSS}{WGSS} \frac{p - K}{K - 1} \tag{6}$$

In (6) K represents the number of clusters, p is the number of pixels in the image, BGSS is the between-group (i.e., inter-cluster) sum of squares and WGSS is the within-group (i.e., intra-cluster) sum of squares. BGSS is made up from the summation of all the possible combinations of the squared distances calculated between two points in different clusters. The computation of WGSS involves the summation of all the possible squared distances in each individual cluster and then adding them up for all the K clusters. Since K is an unknown parameter the VRC was calculated using nine different values from $K = 2$ to $K = 10$ and the one that lead to the maximization of the VRC value was adopted. Maximization requires BGSS (WGSS) to be as greater (smaller) as possible in order to get a better separation of the groups. Since the optimal value of K is image-specific it was found to fluctuate in the 3–7 range for the 2036 input images. Figure 2 shows the way K distributes across the image dataset. There is a peak at $K = 4$ (36% of the cases) followed by $K = 3$ (28% of the cases). In order to use a single value of K the nearest integer from the average over the entire set of input images was established; this value turned out to be $K = 4$. A validation method using bootstrapping was carried out by randomly choosing pixels from the different input images in order to determine a single value of K for the resulting subsample image. This process was replicated several times and the outcome was also $K = 4$ on average.

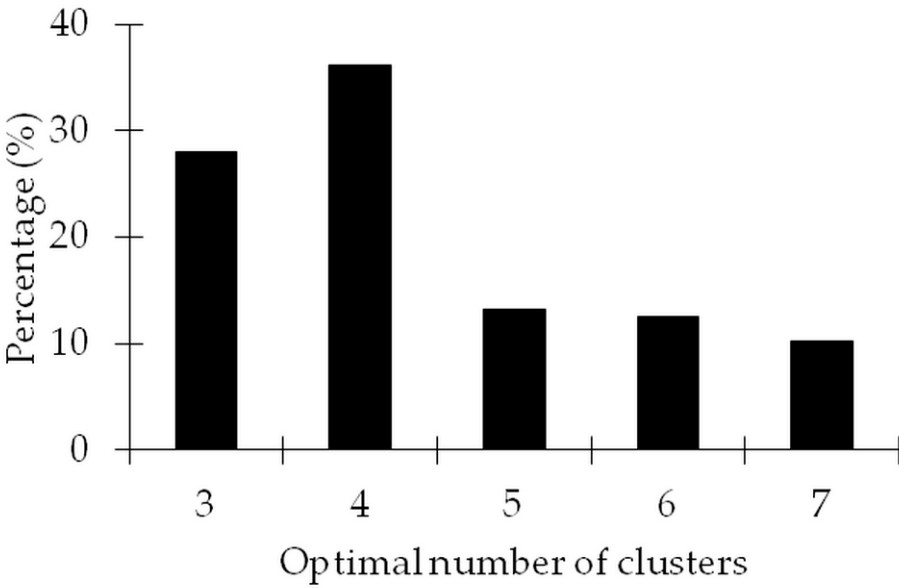

**Figure 2.** Distribution of the optimal number of clusters across the image dataset.

Examples on two different dates whose input images have K = 3 (4 July 2011) and K = 7 (24 April 2013), i.e., the lowest and highest optimal values, respectively, are presented in order to confront their cluster sets with those for K = 4. Individual cluster labeling was carried out over the mean value in an ascending fashion so that cluster 1 has the lowest mean in all sets. Figure 3 shows the input image and clusters 1–3 on 4 July 2011; Figure 4 shows the input image and clusters 1–4 on the same date. Similarly, Figure 5 shows the input image and clusters 1–7 on 24 April 2013 and Figure 6 shows the input image and clusters 1–4 on the same date. Table 1 shows the number of pixels in either set along with the percentage of pixels over the total on these two dates. The only commonality on either date is that the lowest (highest) order cluster includes the coldest (warmest) pixels. The change in the value of K yields a reassignment of pixels between the original and the new clusters. A split will occur when the value of K increases whereas a consolidation will take place when K decreases. On 4 July 2011, the rise in the value of K from 3 to 4 leads to a mild decrease in the number of pixels that populated the original cluster 1 that nets in a gain of pixels for the new cluster 2, the original cluster 2 is partly split between the new clusters 2 and 3, and the original cluster 3 is partly split between the new clusters 3 and 4. Initially, cluster 3 was the most populated cluster, including more than a half of the total number of pixels and representing the largest positive standardized anomalies. Following the splitting the most populated of the new clusters is cluster 3, and the largest positive standardized anomalies in cluster 4 account for a fifth of the total number of pixels. On the other hand, cluster 1 remains the less populated cluster after splitting. On 24 April 2013, the reduction in the value of K from 7 to 4 leads to the transfer of some of the pixels in the original cluster 2 to the new cluster 1, the rest of the pixels in the original cluster 2, the original cluster 3 and a small portion of the original cluster 4 are amalgamated into the new cluster 2, most of the contributions to the new cluster 3 comes from the amalgamation of the original clusters 4 and 5, and the original clusters 6 and 7 are amalgamated into the new cluster 4 with contributions from the original cluster 5 as well. After consolidation cluster 1 remains the less populated cluster and the new cluster 3 is the more populated cluster with half the total number of pixels. The new cluster 4, which includes the largest positive anomalies, represent more than a third of the total number of pixels.

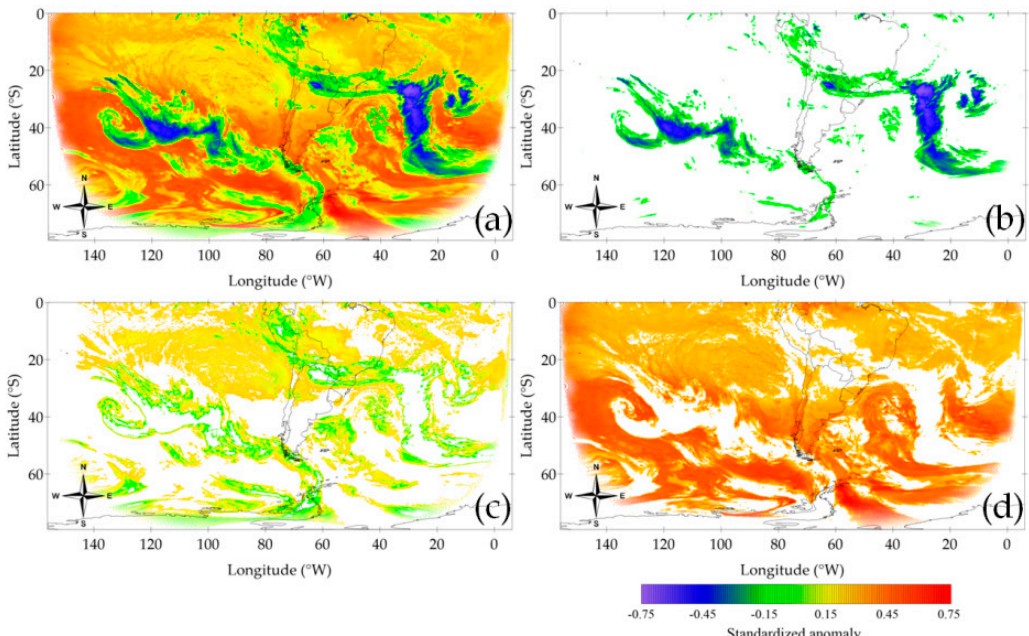

**Figure 3.** Input image (**a**) and clusters 1–3 (**b**–**d**) on 4 July 2011. The color scale is the same for all the panels.

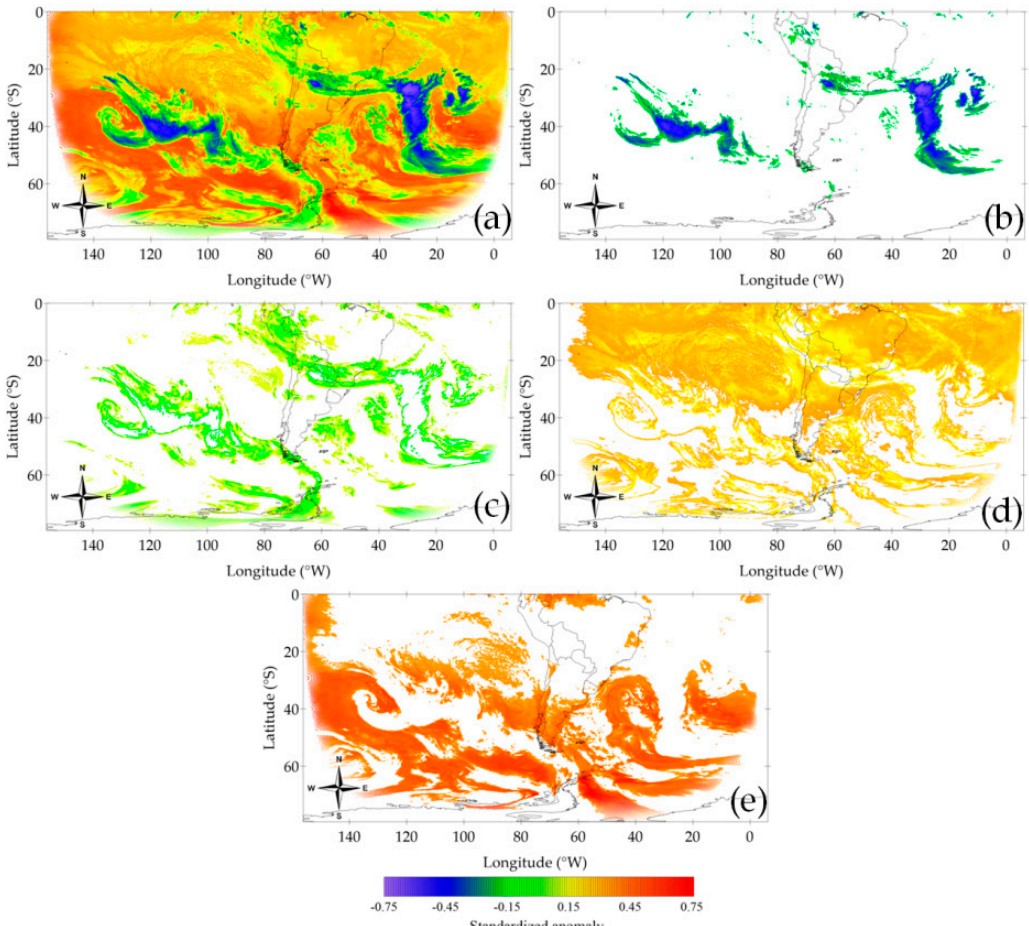

**Figure 4.** Input image (**a**) and clusters 1–4 (**b**–**e**) on 4 July 2011. The color scale is the same for all the panels.

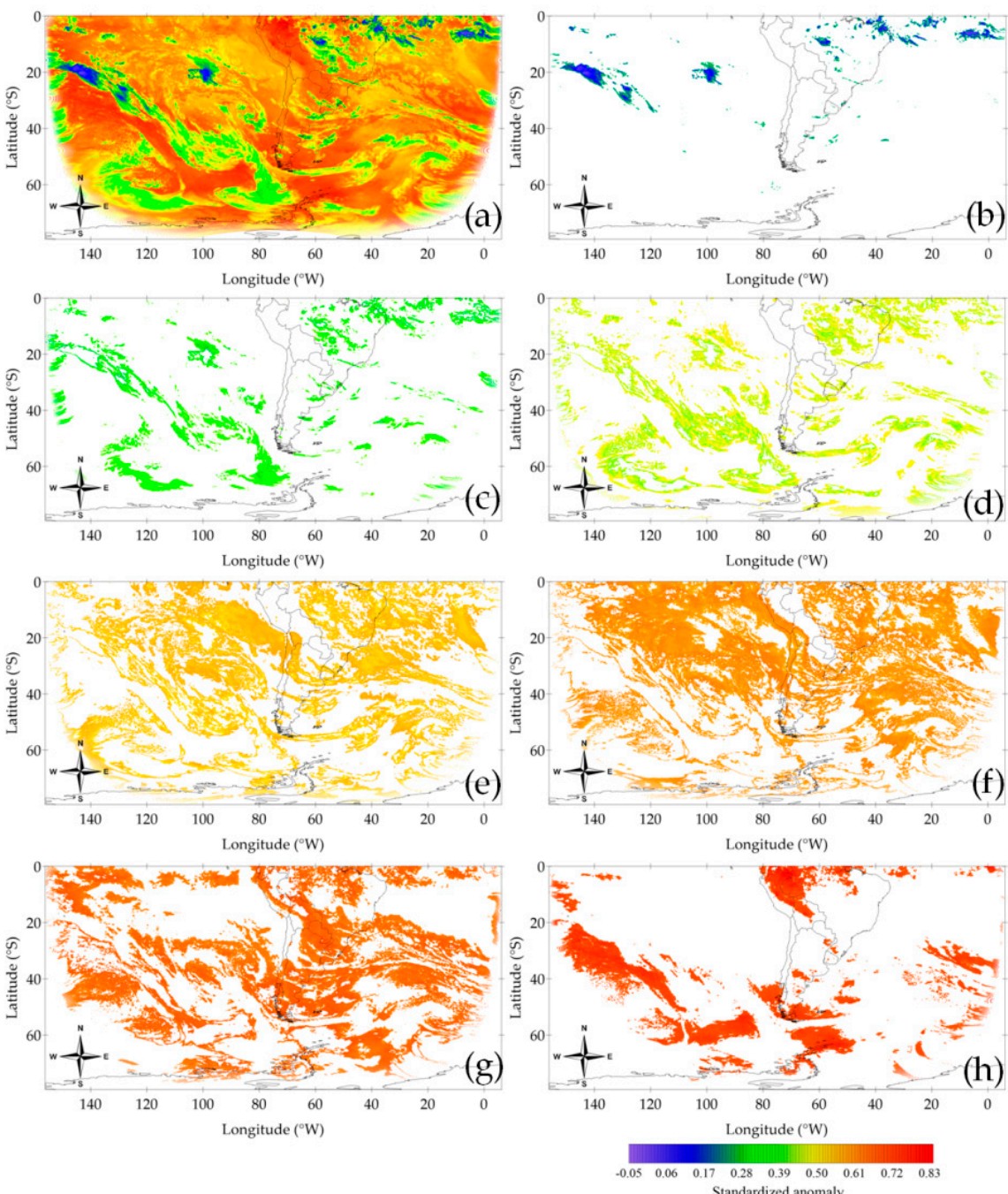

**Figure 5.** Input image (**a**) and clusters 1–7 (**b**–**h**) on 24 April 2013. The color scale is the same for all the panels.

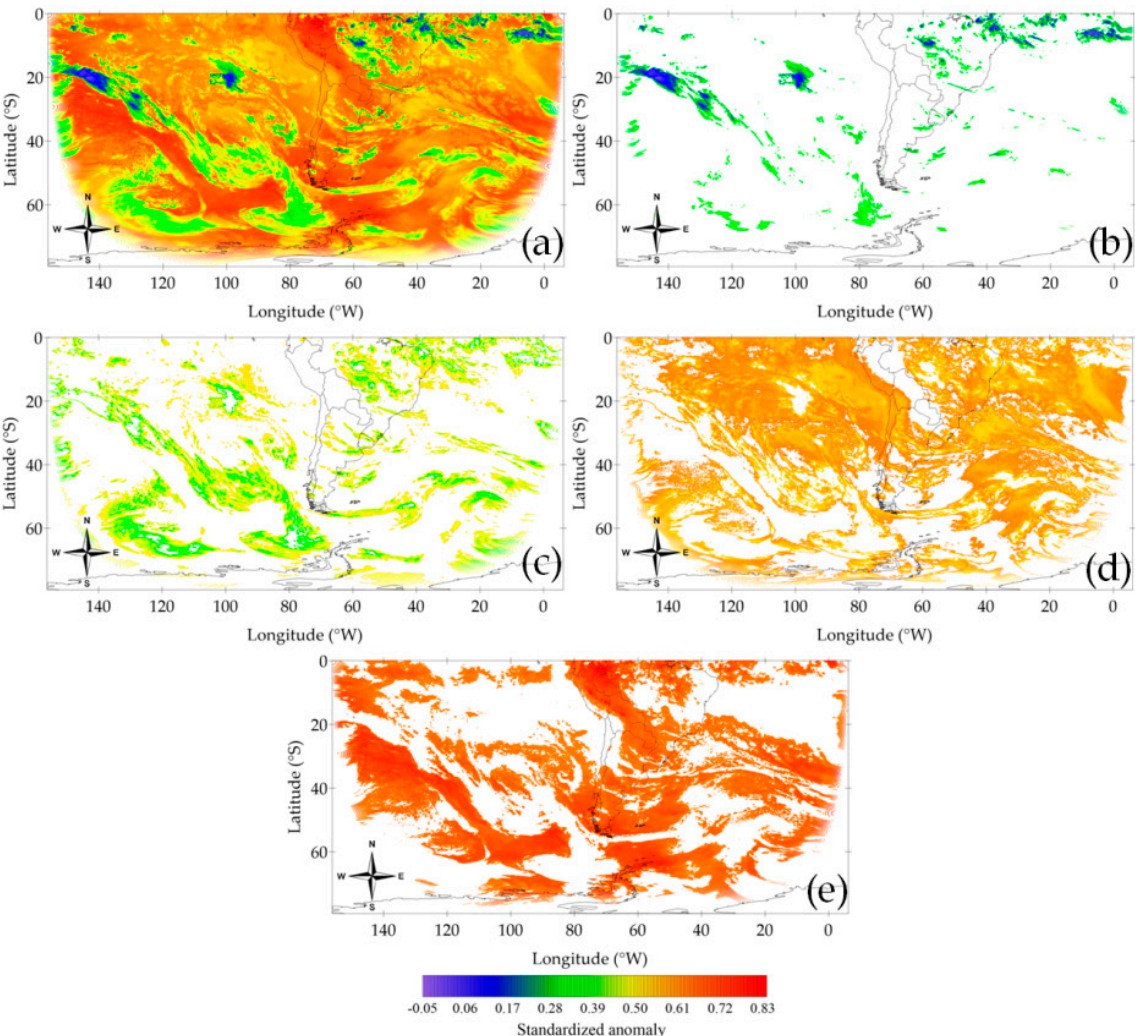

**Figure 6.** Input image (**a**) and clusters 1–4 (**b**–**e**) on 24 April 2013. The color scale is the same for all the panels.

**Table 1.** Number of pixels included in each cluster for two selected dates that have the optimal number of clusters not coinciding with the fixed value of 4 used in the paper. Similar figures for the number of clusters set to 4 are also shown. The representativeness of each individual cluster within its group is presented in round brackets and it is estimated through the percentage over the total number of pixels in each image.

| Cluster # | Date | | | |
|---|---|---|---|---|
| | **4 July 2011** | | **24 April 2013** | |
| 1 | 465,021 (9.22%) | 311,922 (6.18%) | 86,072 (1.71%) | 182,495 (3.62%) |
| 2 | 1,813,436 (35.94%) | 748,304 (14.83%) | 228,091 (4.52%) | 622,275 (12.33%) |
| 3 | 2,767,078 (54.84%) | 2,984,436 (59.15%) | 405,573 (8.04%) | 2,523,610 (50.02%) |
| 4 | | 1,000,873 (19.84%) | 1,092,484 (21.65%) | 1,717,155 (34.03%) |
| 5 | | | 1,549,494 (30.71%) | |
| 6 | | | 1,175,199 (23.29%) | |
| 7 | | | 508,622 (10.08%) | |
| **Total** | 5,045,535 (100.00%) | 5,045,535 (100.00%) | 5,045,535 (100.00%) | 50,455,535 (100.00%) |

The numbers in Table 1 were used to estimate the information entropy [63] as follows

$$S = -\sum_i p_i \ln\left(p_i\right) \qquad (7)$$

In (7) $p_i$ stands for the "probability" of each cluster—estimated as the ratio between the number of pixels in the cluster and the total number of pixels in the input image. Given that the lower the value of the entropy the higher the information provided relation (7) can be used as an absolute value for comparisons between cluster sets regardless of the value of K. With the aid of Table 1 the calculation of the information entropy on 4 July 2011 gave 0.92 and 1.09 for K = 3 and K = 4, respectively. In other words, a fewer number of clusters as estimated by the VRC gave more information. On the other hand, the value of the entropy on 24 April 2013 was 1.68 and 1.09 for K = 7 and K = 4, respectively. This particular case stresses that an optimal number of clusters does not necessarily imply the best distribution of information across them.

### 2.3. Evaluation of Cloud Top Heights

The pixel value in each cluster was multiplied by the corresponding monthly SD and added the corresponding monthly mean in order to present the clusters in the original variable $T_b$. A linear regression between temperature and height, using temperature expressed in Kelvin as the predictor, was conducted over reanalysis data in order to estimate cloud top heights (CTHs) from $T_b$ values. The reanalysis database used for these calculations consisted of air temperatures and geopotential heights retrieved from the National Centers for Environmental Prediction/National Center for Atmospheric Research (NCEP/NCAR) reanalysis webpage (https://www.esrl.noaa.gov/psd/data/gridded/data.ncep.reanalysis.pressure.html) [64]. Each GOES pixel was associated to the closest grid point in the reanalysis. The regression was carried out for the later dataset between 1000 and 70 hPa so that information from both the troposphere and the lowermost stratosphere was considered. This was done for each of the 2036 individual dates and for the total number of 5,045,535 pixels in each image. The regression coefficients were used to associate $T_b$ values with CTHs. Given that the instrument on board the satellite did not collect information from below the Earth surface negative CTHs that resulted from the regression were not included in the analysis.

## 3. Results

The proper way to describe the outcome of the clustering process is to analyze the clusters of each image in the dataset individually. For the sake of brevity, only three sample dates were selected for this purpose. On the other hand, a description of the major features for the full analysis will be made from a seasonal perspective.

### 3.1. Sample Dates

### 3.1.1. Weather Analysis

In order to provide a detailed description of the cluster classification results, three different sample dates were randomly selected. These dates are 1 August 2011 (austral winter), 4 January 2014 (austral summer) and 12 October 2016 (austral spring). Prior to analyzing the configuration of each cluster it is appropriate to describe at least the more prominent features of the CC that can be seen in the original $T_b$ images for these dates. An exhaustive analysis is however beyond the scope of the present work. Figure 7 shows the original $T_b$ monochromatic images on these three sample dates along with the corresponding colored $T_b$ values. It also shows the CTHs on 1 August 2011.

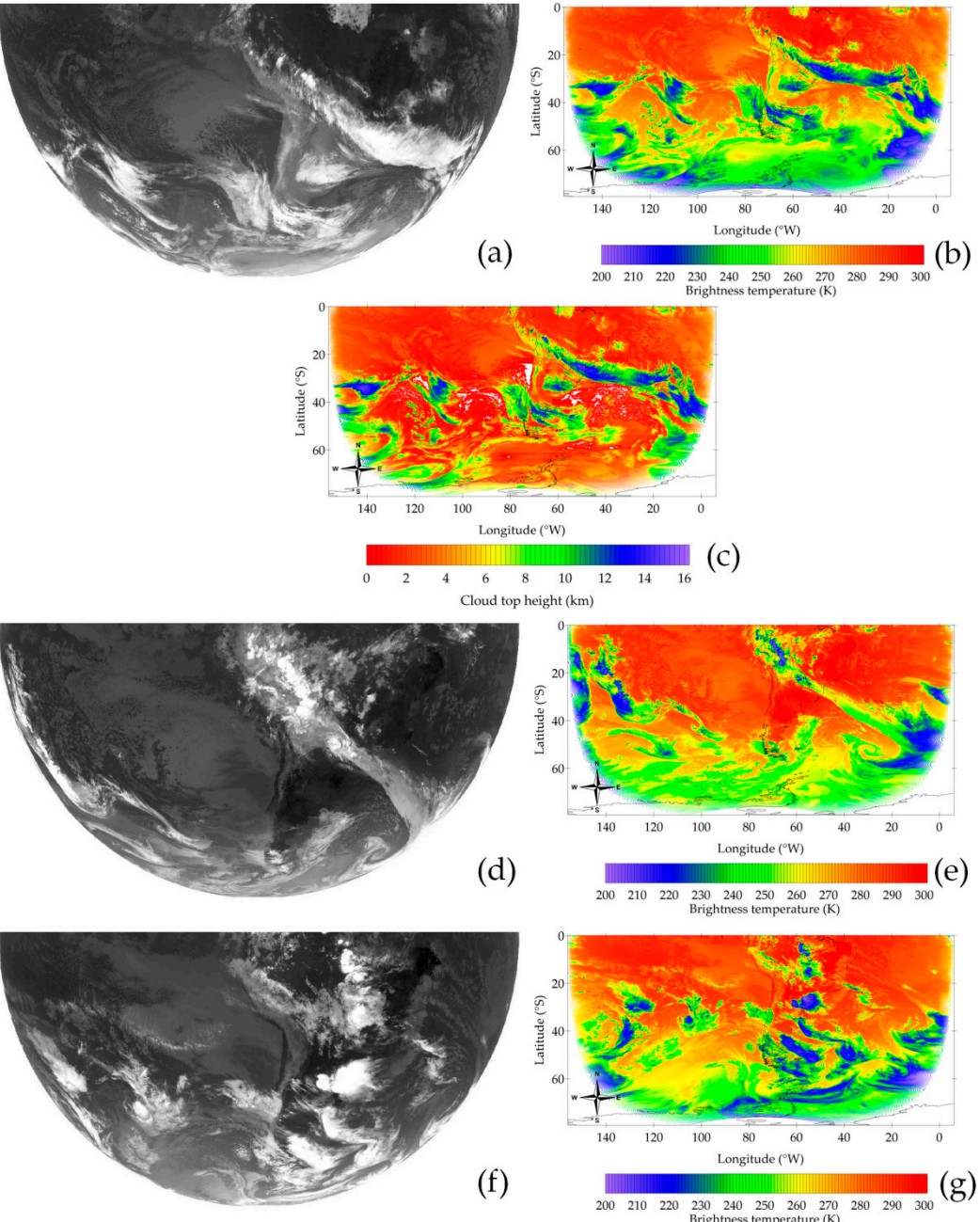

**Figure 7.** Monochromatic brightness temperature ($T_b$) as calculated from the original IR images (left) and colored $T_b$ values in Mercator projection (right) for: (**a–c**) 1 August 2011, (**d,e**) 4 January 2014 and (**f,g**) 12 October 2016. Plate (**c**) shows the cloud top heights (CTHs) that correspond to Figure 2a,b. Blank CTH pixels that do not match their colored $T_b$ counterparts are negative and were excluded. See text for further details.

The poleward edge of extensive shields of cirrus clouds (Cis) generally outlines the anticyclonic curvature of the upper tropospheric jet (UTJ) [20]. Such a structure is clearly visible in Figure 7a with Cis in tones corresponding to colder $T_b$ values approximately located over Southern Brazil and the Atlantic. As shown in Figure 7b, these clouds have $T_b$ values of approximately 210–220K in the area, in correspondence with CTHs close to 14 km (Figure 7c), and in turn in agreement with the location of the tropical tropopause in connection with the so-called upper tropospheric "subtropical front" found at these latitudes [65]. Mounted on a mid-latitude wavetrain, a frontal zone can be easily identified over the southern tip of SA, and there are visible signs of a cut-off circulation north of this front over central

Argentina. Middle-to-high clouds seem to dominate in both of these two cloud systems. Even though Southern SA is under the influence of frontal activity all year round [66], generally speaking winter fronts are typically stronger due to an equatorward advance of baroclinicity, whose main driver is related to a strengthening of UTJs during the season across the entire SH [67]. Frequently associated to cyclogenesis [68], UTJs are more generally connected to the maintenance of the storm tracks due to their baroclinicity [69]. Figure S2 shows the geopotential height at ten different levels from 850 (lower troposphere) to 100 hPa (lower stratosphere) along with the wind field and the kinetic energy on 1 August 2011. It shows the presence of an UTJ, with maximum strength at 200 hPa in the leading part of an upper-level trough that is in a "tear-off" stage over central Argentina.

In lighter grey tones in Figure 7d, another frontal structure in a northwest-by-southeast position can be seen as a cloud band stretching from the east of the Ande—where several convective systems can be seen in the foreground—towards the South Atlantic. The $T_b$ values of the convective cells located in North-Western Bolivia and Eastern Brazil were around or below 210 K; convective-related clouds were also visible over Paraguay (Figure 7e). This cloud arrangement is characteristic of summer and is linked to an area of moisture flux convergence [70]. In fact, the region is associated to the local minima of outgoing longwave radiation where the well-known South Atlantic convergence zone (SACZ) develops as a result of the cold fronts reaching the region becoming stationary there ([71] and references therein). The south-easternmost tip of the cloud structure that sits along the SACZ spirals cyclonically (i.e., clockwise in the SH) well over the South Atlantic and bears a strong resemblance with an occlusion in line with the circulation features of the region [66].

The set of Figure 7f,g seems to have the largest number of distinct clouds between the three presented dates. Among these clouds, there were two contiguous mesoscale convective complexes (MCCs) situated in the eastern portion of central SA, approximately over Northern Uruguay and Southern Brazil, a region where MCCs tend to develop [72]. With an even texture, these very high clouds are characterized by lighter grey tones in both systems in Figure 7f, while Figure 7g shows that the $T_b$ values found at the top of these two MCCs were inclined towards the lowest values of the scale. The western, round-shaped MCC was smaller than the eastern one, from which cirrus plumes can be seen streaming off in its north-eastern portion following the westerly flow aloft.

3.1.2. Cluster Analysis

Figure 8 shows the configuration of clusters 1–4 on the three selected dates. Even though pixels represented by the lowest portion of the scale are found in each of the clusters their number generally decreased in favor of hotter pixels as the cluster order increased. Figure 9 shows the silhouette diagrams for these case studies. In general, each pixel was well matched to their own cluster. The information entropy for these dates was 0.96 (01/08/2011), 1.10 (04/01/2014) and 1.14 (12/10/2016; results not shown). These results mean that between the palettes of clouds present in these three days (cf. Figure 7) a better representation by means of a four-cluster set was achieved for the first of the dates. Another interpretation is that in terms of cloudiness the last date is the one that is more disordered.

According to the number of pixels in Figure 8a–c, cluster 1 accounts for approximately 8%, 4% and 7%, respectively, of the corresponding original $T_b$ images (not shown). Unlike with the rest of the clusters, pixels in cluster 1 on the selected dates did not extend beyond approximately 60 °S. Table 2 shows the mean values and the SDs for both $T_b$ and CTH for all the clusters in Figure 8. The mean $T_b$ value for Figure 8a–c was 232 K, 230 K and 228 K, respectively; SD equaled 11 K, 12 K and 11 K, respectively. In terms of CTH, the mean values were 10.71 km, 11.56 km and 10.26 km, respectively, and the SDs were 1.79 km, 2.01 km and 2.75 km, respectively. The distributions of $T_b$ and CTH for the clusters in Figure 8 are shown in Figure 10. Regarding cluster 1, Figure 10a shows that the $T_b$ pixels spanned the support's lower half, with values below approximately 260 K. The distributions in Figure 10b were more scattered than those in Figure 10a, highlighting the fact that the same temperature may be related to distinct heights at different regions. The Cis associated to the UTJ's curvature and the

upper-level trough in Figure 7b or the convective cells over Bolivia, Brazil and Paraguay in Figure 7d were just examples of the CC represented by this cluster.

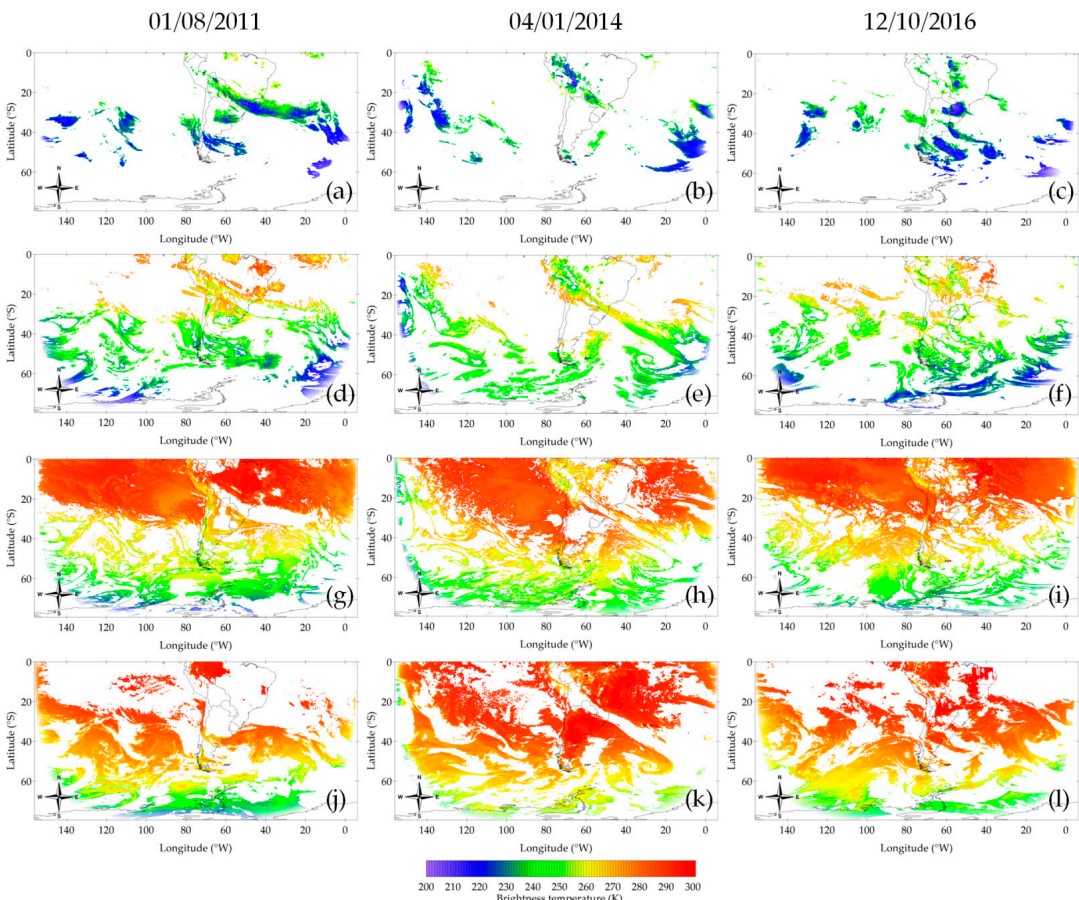

**Figure 8.** Pixel patterns for 1 August 2011 (left), 4 January 2014 (centre) and 12 October 2016 (right). Clusters 1, 2, 3 and 4 are shown in plates (**a**–**c**), (**d**–**f**), (**g**–**i**) and (**j**–**l**), respectively. The color scale is the same for all the panels.

**Table 2.** Brightness temperature $(T_b)$ and cloud top height (CTH) mean values and standard deviations (SDs) for clusters (#) 1–4 on the sample dates.

| Date | $T_b$ (K) Mean (SD) | | | | CTH (km) Mean (SD) | | | |
|---|---|---|---|---|---|---|---|---|
| | #1 | #2 | #3 | #4 | #1 | #2 | #3 | #4 |
| 01/08/2011 | 232 (11) | 256 (15) | 278 (13) | 278 (12) | 10.71 (1.79) | 6.04 (1.91) | 2.16 (1.15) | 1.22 (0.92) |
| 04/01/2014 | 230 (12) | 253 (10) | 280 (10) | 285 (9) | 11.56 (2.01) | 7.34 (1.63) | 2.94 (1.39) | 1.61 (1.00) |
| 12/10/2016 | 228 (11) | 253 (13) | 279 (11) | 284 (11) | 10.26 (2.75) | 6.06 (2.67) | 2.98 (1.28) | 1.44 (0.84) |

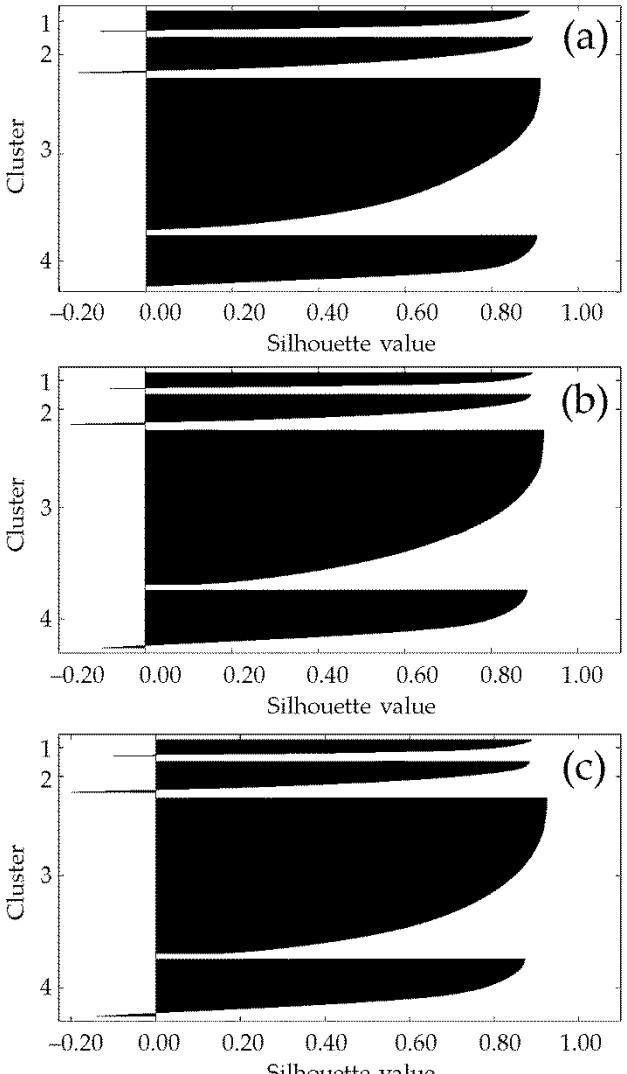

**Figure 9.** Silhouette diagrams for the clusters shown in Figure 8: (**a**) 1 August 2011, (**b**) 4 January 2014 and (**c**) 12 October 2016.

Cluster 2's number of pixels in Figure 8d–f represents 14%, 11% and 12% of the original $T_b$ images, respectively (not shown). The mean $T_b$ values for Figure 8d–f were 256 K, 253 K and 253 K, respectively, while the SDs were 15 K, 10 K and 13 K, respectively (Table 2). As to CTH, the mean values were 6.04 km, 7.34 km and 6.06 km, respectively, and the SD values were 1.91 km, 1.63 km and 2.67 km, respectively. In line with the aforementioned results Figure 10c (10d) shows that the $T_b$ (CTH) distribution was shifted towards greater (lower) values of the corresponding scales. Figure 8d–f shows that pixels in cluster 2 spanned the entire domain. Nevertheless, the abundances for the colder pixels seemed to be greater for latitudes beyond 30 °S. Cloud decks that surround cluster 1's higher clouds in the region of the UTJ-associated Cis (cf. Figure 7a,b) were well represented by this cluster; most of the pixels along the SACZ and portions of the occlusion in the Atlantic (cf. Figure 7c,d) also were.

Cluster 3's number of pixels in Figure 8g–i dramatically increased with respect to the former two clusters to account for 58%, 43% and 52%, respectively, of the total number of pixels in the original $T_b$ images (not shown). Pixels in these images spanned the entire domain. The mean $T_b$ values for Figure 8g–i were 278 K, 280 K and 279 K, while the SD was 13 K, 10 K and 11 K, respectively (Table 2). In terms of CTH, the mean values were 2.16 km, 2.94 km and 2.98 km and the SD equaled 1.15 km, 1.39 km and 1.28 km, respectively. The $T_b$ and the CTH distributions (Figure 10e,f) were in agreement with these results. Moreover, these two histograms were sharper than the previous ones. As with

cluster 2, the colder pixels in Figure 8g–i spanned the domain's lower portion. A feature shared by these three figures was the presence of yellow-red pixels both off the Chilean and the Peruvian coasts and off the Angolan and the Namibian coasts (the African continent was out of sight in all the panels of Figure 8). These pixels were located in regions where persistent stratocumulus (Scs) occurs [73,74]. Scs are usually found below 2000 m independent of the Earth's region (WMO 2019). Some of the pixels that wrap around the occlusion in the Atlantic (cf. Figure 7d) were included in this cluster (Figure 8h).

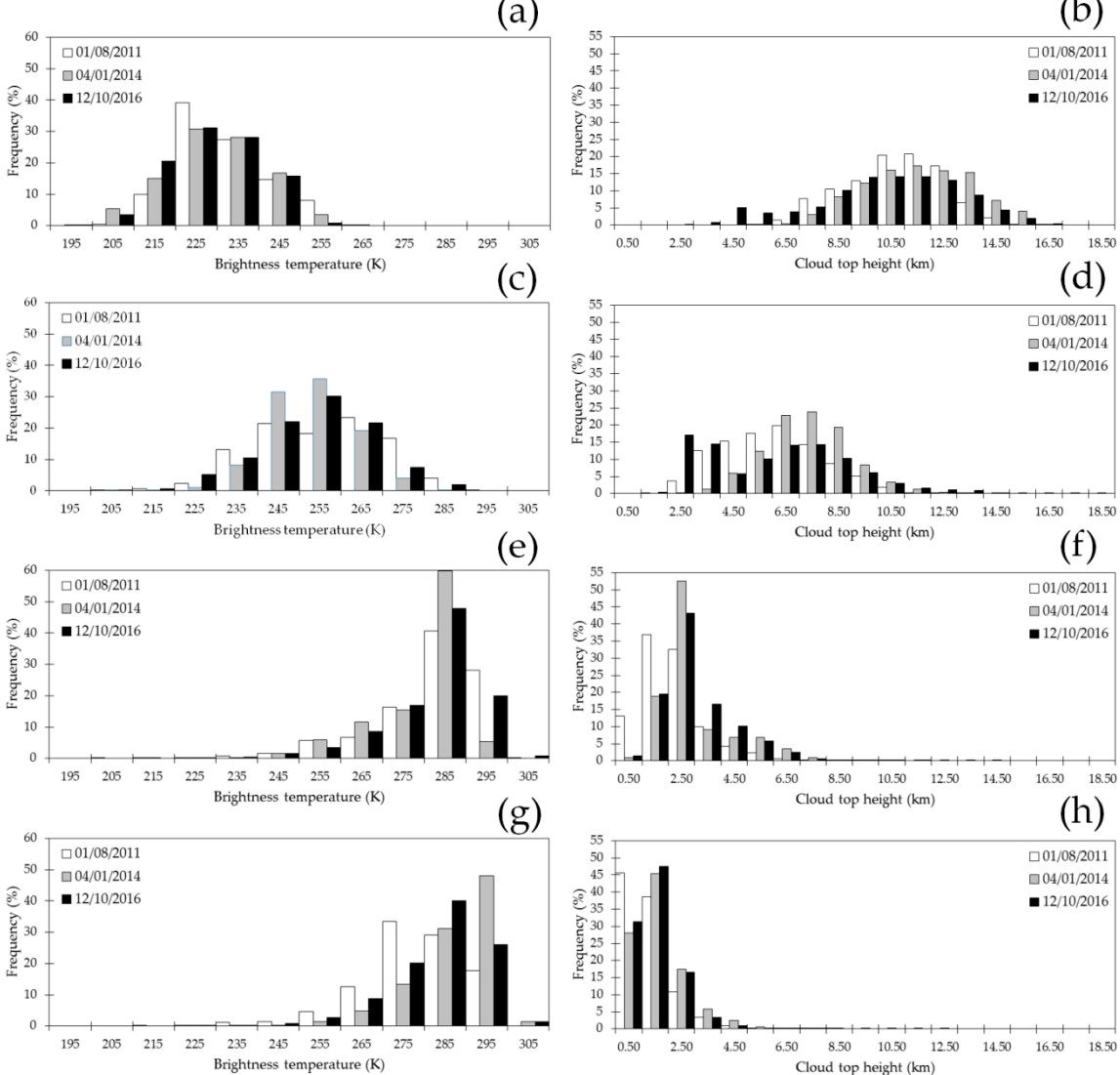

**Figure 10.** Frequency distributions on the selected dates for (**a**,**b**) cluster 1; (**c**,**d**) cluster 2; (**e**,**f**) cluster 3 and (**g**,**h**) cluster 4.

Figure 8j–l shows the configuration of cluster 4 on the selected dates. The number of pixels represented in them accounted for 20%, 42% and 29%, respectively, of the total number of pixels in the input images (not shown). Aside from Figure 8k, these percentages marked a difference in that they included fewer pixels than cluster 3. As with cluster 2 and 3, all latitudes were spanned by cluster 4's pixels on the selected dates. Reddish pixels were by far the most abundant in all three cases. The $T_b$ mean value for Figure 8j–l was 278 K, 285 K and 284 K, respectively, while the SD values were 12 K, 9 K and 11 K, respectively. As to CTH, the mean values were 1.22 km, 1.61 km and 1.44 km, respectively; the SD values equaled 0.92 km, 1 km and 0.84 km, respectively (Table 2). The histograms of $T_b$ for clusters 3 and 4 bore a resemblance both in the shape and in the support. By contrast, the CTH

cluster 4's distributions shifted towards the lowest values, with the class intervals that had a noticeable contribution extending through 5 km (Figure 10h).

### 3.2. Seasonal Features

The first part of this subsection is dedicated to the analysis of the mean values, SDs and frequency distributions from a seasonal perspective. Table 3 shows the seasonal grand-time mean values and SDs for both $T_b$ and CTH. The term grand-time mean was used here to reflect that the calculations were carried out within the entire period of analysis irrespective of the year, incorporating information from December, January and February (summer), March, April and May (autumn), June, July and August (winter), and September, October and November (spring). A general result from Table 3 is that the seasonal grand-time mean values for $T_b$ (CTH) increased (decreased) as the cluster order increased, and that the intra-seasonal variability decreased as the cluster order increased for both $T_b$ and CTH. With the exception of cluster 4, in terms of CTH winter presents the seasonal lowest mean values and autumn had the highest ones; this made the autumn-to-winter transition generally steeper than in any other season-to-season case. As to variability, it was greater in spring for clusters 1 and 2, in autumn for cluster 3 and in summer for cluster 4. The grand-time histograms for the four seasons are presented in Figure 11. They did not differ in a remarkable fashion from the corresponding ones shown in Figure 10.

**Table 3.** Seasonal brightness temperature ($T_b$) grand-time means and standard deviations (SDs, in round brackets) for clusters (#) 1–4.

| Season | $T_b$ (K) Mean (SD) | | | | CTH (km) Mean (SD) | | | |
|---|---|---|---|---|---|---|---|---|
| | #1 | #2 | #3 | #4 | #1 | #2 | #3 | #4 |
| Summer | 233 (10) | 259 (7) | 280 (5) | 284 (4) | 11.07 (1.66) | 6.37 (1.28) | 2.68 (0.69) | 1.64 (0.57) |
| Autumn | 230 (9) | 255 (8) | 280 (6) | 284 (5) | 11.34 (1.52) | 6.82 (1.35) | 2.70 (0.72) | 1.61 (0.55) |
| Winter | 232 (10) | 261 (10) | 279 (5) | 279 (6) | 10.43 (1.51) | 5.39 (1.27) | 2.06 (0.57) | 1.40 (0.54) |
| Spring | 231 (10) | 256 (9) | 279 (5) | 284 (6) | 10.47 (2.06) | 5.98 (1.38) | 2.56 (0.67) | 1.45 (0.49) |

The seasonal character of the clusters represented in the columns of Table 3 warrants them to be termed "cloud regimes" to parallel similar plots that are present in the literature [75,76]. Clouds in cluster 1 represent high clouds including, but not restricted to, cirriform clouds, as this cluster is likely capturing the features of clouds that have a vertical extent as well, such as the cumulonimbus (Cbs). This is the less dominant cluster, representing, on average, 6% of the pixels. On the opposite side of the classification clouds in cluster 4 have their tops below 2 km. According to [77] they are classified as stratiform clouds. This is the second most dominant cluster as it represents, on average, 32% of the pixels. Clusters 2 (49%) and 3 (13%) spanned the middle troposphere. Altostratus and nimbostratus are examples of clouds that may populate them [77].

Given that seasonality implies that a number of different physical processes conspire to give the figures shown in Table 3 and Figure 11, it is difficult to draw conclusions about individual processes from these aggregated values. Monthly means were calculated for both the number of pixels that populate each cluster and the entropy. These values are shown in Figure 12. A noticeable feature in this figure is that the entropy closely followed the evolution of cluster 4. It can also be seen that the entropy increased when the number of pixels in clusters 3 and 4 were close to one another, i.e., when these two clusters became less distinguishable from each other. This tended to occur in fall and winter. In an endeavor to provide further insights on the intraseasonal variability and to shed light on the possible physical processes involved the mean values in Figure 12 were correlated with several circulation indices both on a grand-time seasonal basis and for the entire period, i.e., disregarding season or year.

The calculations were carried out using the Spearman's correlation coefficient ($\rho$), which acts over the ranks of the pair of correlated variables (rather than over the raw values) and permits establishing whether there exists a relationship between the variables by means of a monotonic increasing or decreasing function [78]. It is worth noting that the method did not disclose the specific functional relationship between the variables, it just assessed the degree of monotonicity. The circulation indices that were chosen for this specific purpose were the Antarctic Oscillation the Dipole Mode Index (DMI) [79], the Madden–Julian Oscillation (MJO) [80] at ten different longitudes, the Pacific Decadal Oscillation (PDO) [81], the Quasi-Biennial Oscillation (QBO) [82] and the Southern Oscillation Index (SOI) [83]. The values of $\rho$ are presented in Table 4, which only included significant values. The DMI was excluded from the table as the $\rho$ values resulted as not significant in all cases.

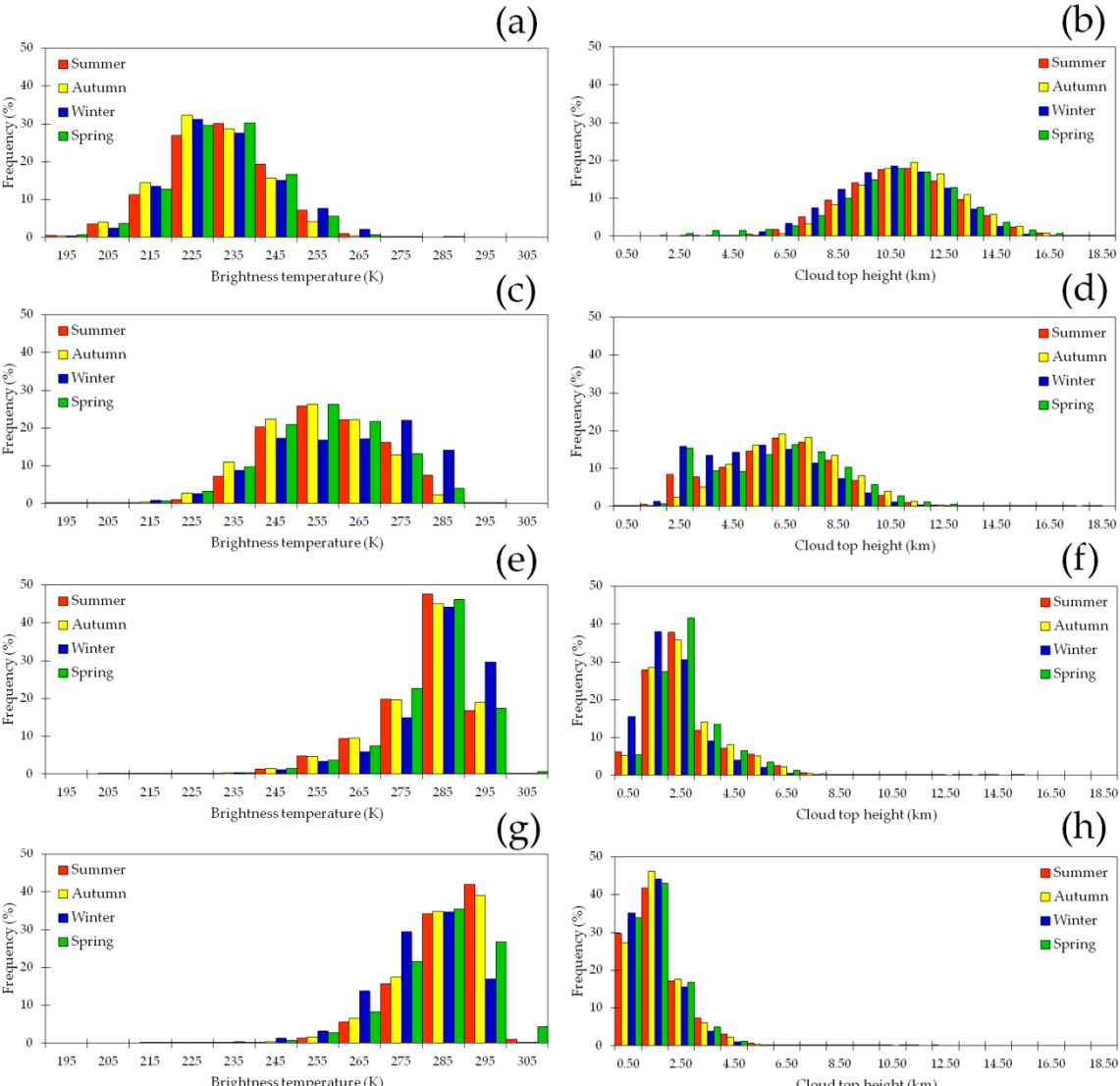

**Figure 11.** As in Figure 10 but for the seasonal frequency distributions. These distributions were prepared including the entire set of 2036 analyzable images.

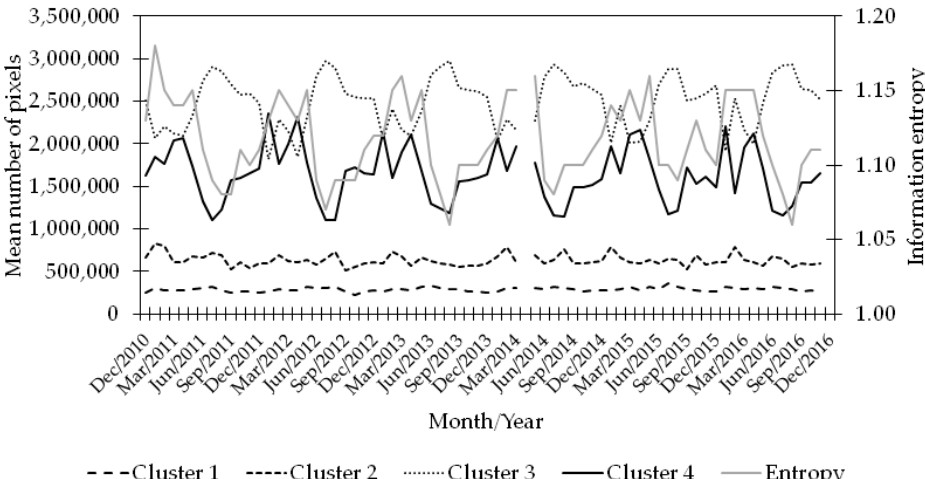

**Figure 12.** Monthly mean values for the number of pixels that populate each cluster and for the entropy.

The AAO, otherwise known as the Southern Annular Mode (SAM) [84], has several definitions depending upon the database it is constructed from ([85] and references therein). The AAO monthly time series used here is constructed as the projection of the monthly 700 hPa height field south of 20 °S onto the leading pattern that results from an Empirical Orthogonal Function analysis using the 700 hPa height over the 1979–2000 period [86]. The leading pattern consists of two components at two different latitudes, namely an arrangement of three centers of action along 45 °S with one of them extending over Southern SA and the South Atlantic, and anomalies of the opposite sign over Antarctica [87]. The AAO has significant correlations with cluster 2 and with the entropy, in both cases with no signals in any particular season but year-long. The SAM's presence in the troposphere all through the year [88] may explain the reasons why correlations are only significant year-round. The positive ρ values means that there exists a certain degree of direct association between the AAO signal in cluster 2's number of pixels and in the entropy. The increase (decrease) in the entropy was partly explained by a positive (negative) phase of the AAO. Dynamically, this may be related to the SAM being associated with synoptic transients [89], which tend to create a more disordered troposphere. Following this reasoning, the AAO modulation of cluster 2's number of pixels may come from the fact that this cluster represents cloudiness related to synoptic processes, i.e., located in the middle-to-upper troposphere (cf. Table 3). No further details regarding this will be provided since the topic was beyond the scope of this paper and a number of the processes involved are still under investigation by the scientific community.

The negative correlation between cluster 1's number of pixels and the SOI all through the year is in compliance with an increase in the number of mesoscale convective systems (MCCs) in the region during El Niño events [90]. The existence of a direct linear relationship between the PDO and the El Niño/Southern Oscillation (ENSO) [91] can be recalled in order to interpret the correlations with both the PDO and the SOI. Negative (positive) values of the SOI are indicative of El Niño (La Niña) events. The PDO correlates with the entropy all through the year. Since the PDO is a year-round phenomenon [91] a relationship that partly reflects a direct modulation of the entropy on a year-round basis is not surprising, especially considering the aforesaid PDO/ENSO connection and recalling the tendency of an ENSO event to have a more disordered troposphere through the increase in the global temperature [92]. In spite of the strength of the PDO/ENSO association no entropy/SOI significant correlations were found on a year-round basis. Table 4 also shows that the positive (negative) phase of the PDO partly modulated a spring-time increase (decrease) in both the number of pixels associated to convective clouds and the entropy in the study region. These correlations were twice as strong as the year-round ones and this in concordance with the PDO/ENSO relationship being stronger during spring [91]. By means of the PDO/ENSO relationship the remaining correlations with the SOI have already been dealt with.

**Table 4.** Spearman's correlation coefficients between the monthly mean time series of the number of pixels in each of the clusters or of the entropy and the monthly time series of the Antarctic Oscillation (AAO), the Pacific Decadal Oscillation (PDO), the Quasi-Biennial Oscillation (QBO), the Southern Oscillation Index (SOI) and the Madden–Julian Oscillation (MJO) at ten different longitudes. Correlations were calculated seasonally (December/January/February (DJF), March/April/May (MAM), June/July/August (JJA) and September/October/November (SON)) and for the entire period, i.e., disregarding year or season. Values shown are significant at a 95% confidence level.

| Cluster/ Entropy | Season | AAO | PDO | QBO | SOI | MJO | | | | | | | | | |
| --- | --- | --- | --- | --- | --- | --- | --- | --- | --- | --- | --- | --- | --- | --- | --- |
| | | | | | | 20°E | 70°E | 80°E | 100°E | 120°E | 140°E | 160°E | 120°W | 40°W | 10°W |
| #1 | All | | | | −0.27 | −0.53 | −0.37 | | 0.42 | 0.46 | 0.53 | 0.46 | | −0.30 | −0.50 |
| | DJF | | | | | −0.74 | −0.72 | | | 0.53 | 0.75 | 0.78 | | | −0.62 |
| | MAM | | | | | −0.63 | | | 0.51 | 0.57 | 0.61 | 0.51 | | | −0.56 |
| | JJA | | | | | | | | | | | | | | |
| | SON | | 0.50 | 0.50 | −0.52 | | | | | | | | | | |
| | All | 0.24 | | | | −0.31 | −0.25 | | | | | 0.31 | 0.29 | | −0.27 |
| | DJF | | | | | | | | | | | | | | |
| | MAM | | | | | | | | | | | | | | |
| | JJA | | | | | | | −0.51 | | | | | | | |
| | SON | | | | | | | | | | | | 0.47 | | |
| #3 | All | | | | | | | | | | | | | | |
| | DJF | | | | | | | | | | | | | | |
| | MAM | | | | | | | | | | | | | | |
| | JJA | | | | | | | | | | | | | | |
| | SON | | | | | | 0.49 | | | | | | | | |
| | All | | | | | | | | | | | | | | |
| | DJF | | | | | | | | | | | | | | |
| | MAM | | | | | | | | | | | | | | |
| | JJA | | | | | | | 0.48 | | | | | | | |
| | SON | | | | | | | | | | | | | | |
| Entropy | All | 0.25 | 0.25 | | | | | | | 0.24 | | | | | −0.24 |
| | DJF | | | | | −0.60 | −0.71 | | | | 0.54 | 0.76 | | | |
| | MAM | | | | | −0.74 | −0.49 | | 0.53 | 0.64 | 0.72 | 0.64 | | | −0.66 |
| | JJA | | | | | | | | | | | | | | |
| | SON | | 0.59 | 0.51 | −0.52 | | −0.49 | | | | | 0.52 | | | |

The QBO index used here consisted of a monthly zonal average of the equatorial zonal winds at 30 hPa. It is therefore the only index that includes a stratospheric meteorological variable. With the exception of the correlation with the entropy all-year long, correlations with the QBO matched the ones with the PDO both in value and in season, i.e., during spring there was an increase both in cluster 1's number of pixels and in the entropy with "positive" phases of the QBO (i.e., weak easterlies of westerlies). The results presented here are in agreement with those in [93] who observed that regionally the greatest number of pixels with significant correlations between the ultraviolet reflectivity and the QBO occur during spring. The greatest number of MCCs in South America occurs in spring too [90]. This can be partly understood by dint of the QBO/ENSO phasing [82].

The MJO quantifies the intraseasonal variability in the tropical atmosphere and is known for influencing both the tropical and the extratropical weather patterns. More specifically, the MJO plays a role in modulating the precipitation regime through the interaction between the general circulation with deep convection ([80] and references therein). In particular, the active phase of the MJO had an effect in the activation of large convective cloud systems with temperatures at their top in agreement with the ones in the lower portion of the histograms in Figure 11 ([80] and references therein). The correlations between the MJO and cluster 1's number of pixels occurred during (austral) summer and fall—when the MJO is at its peak [80]— and all through the year (cf. Table 4). Even though no significant ρ values were found for the rest of the seasons, it is likely that the significant year-round values are strongly influenced by the contributions from summer and fall since it is in these two quarters when convection is stronger in the SH due to the astronomical forcing. The zonal differences in both the strength and the sign of the correlations were related to the MJO's alternation between its active and suppressed phases across the latitudes it was calculated at, making the strongest positive (negative) ρ values to take place at 140 °E (20 °E). As with cluster 1, the phasing of the MJO can be recalled to explain the year-round correlations with cluster 2's number of pixels across the different longitudes as well, yet the MJO's role in modulating the number of pixels in this cluster was not as strong as with cluster 1's (i.e., the ρ values were lower). On the other hand, there was a seemingly stronger modulation in cluster 2's number of pixels in winter (spring) when convection was suppressed (active) at 80 °E (120 °W). In view of the general agreement of cluster 2 with the cloudiness present in synoptic-scale processes (cf. Figure 12) and their pixels spreading across the entire study region (cf. Figure 8), these results show an apparent connection between the mesoscale and the synoptic scale. Such a relationship between these two different atmospheric scales has been documented for the region (e.g., [94]). The MJO at 70 °E was the only index that correlated with clusters 3 and 4, in spring and winter, respectively. This could be related to the aforementioned swapping in the number of pixels between these two clusters. On the other hand, the entropy in summer and fall seemed to generally decrease when convection was enhanced over Africa (MJO at 20 °E) and the Atlantic and the Indian Oceans (MJO at 10 °W and 70 °E, respectively), and to increase when convective activity at the Tropical Warm Pool (TWP; MJO at 100°E, 120°E, 140°E and 160°E) was at its top. Something similar occurred in springtime and year-round but more restricted longitudinally respecting the MJO.

### 3.3. Identification of Cirrus and Cumulonimbus in Cluster 1

Cirrocumulus (Ccs), cirrostratus (Css), Cbs and Cis populate the upper troposphere [77]. Given that cluster 1 had the lowest $T_b$ values all these types of clouds might coexist on particular dates. These clouds have different properties concerning precipitation so it is desirable to separate Cbs from the rest of them in order to have them isolated given the potential effects these particular clouds have on weather (e.g., flash floods). In a first attempt to do this using the information extracted from the cluster analysis we made use of the area–perimeter fractal dimension as follows. For each individual cluster 1 image different objects, each one of them with clear boundaries that enabled its distinction from the rest of the objects in the image, were identified. The perimeter P of each object was calculated by counting the outermost pixels. Similarly, the area A of each object was estimated by summing up the number of pixels encompassing it. Following [95] objects with A < 200 km$^2$ were excluded from the

analysis. Objects with hollow regions were excluded as well. For each object that fulfills the condition $A > 200$ km$^2$ a quantity R that measures its raggedness was constructed by taking into account the area–perimeter ratio in the following way

$$R = \frac{\ln(P)}{\ln(A)} \tag{8}$$

Expression (8) exploits the notion of fractal dimension as the object's raggedness increases as R increases. A vector including the R parameters was built. The same k-means/k-means++ clustering algorithm described in Section 2.2 was implemented with K = 2 to break this new vector down into two vectors with quite different properties regarding the geometry, one of them capturing the features of Cis (Ci-vector) and the other of Cbs (Cb-vector). For each object in these two vectors the perimeter fractal dimension D(P) can be estimated through the relation ([95] and references therein)

$$P = CA^{\frac{D(P)}{2}} \tag{9}$$

In the above equation C is a constant. The linearization of (9) leads to

$$\ln P = \ln C + \frac{D(P)}{2} \ln A \tag{10}$$

from which both the intercept $\ln C$ and the slope $D(P)/2$ can be easily obtained from a linear fit considering all the individual objects in each Ci- and Cb-vector. By averaging over the 2036 realizations we get $D_{Ci}(P) = 1.178 \pm 0.014$. The second term is the SD across the 2036 Ci-vectors. In a similar fashion, $D_{Cb}(P) = 1.364 \pm 0.022$. These values, which are in very good agreement with the ones in [95], were used to classify the pixels in cluster 1.

Figure 13a shows the breakdown of Figure 8a into groups of pixels of Cis and Cbs. In this particular case Cis and Cbs represent approximately 89% and 9% of cluster 1's total number of pixels, respectively. The remaining 2% of the pixels correspond to the clouds that were unaccounted for by the condition $A > 200$ km$^2$. The largest cluster seen in Figure 13a was the one associated to Cis that outline the anticyclonic curvature of the UTJ discussed in Section 3.1.1. The quarterly time series of mean seasonal percentages of Cis and Cbs across the study period are shown in Figure 13b. Cluster 1's population of Ci and Cb pixels across all seasons is, on average, around 83% and 6%, respectively. In agreement with the existing literature [96] the percentages of Ci pixels reached a maximum (minimum) in winter (summer). These results are in accordance with a boreal lower temperature in the troposphere that favored both the formation and the maintenance of ice crystals, which are the constituents of this type of clouds [97]. A concomitance in the increase or decrease in the number of Ci and Cb pixels is expected seasonally. This is a manifestation of one of the formation processes of Cis as they are known to outflow from convective Cbs (i.e., "anvil clouds" [98] (p. 44)). Upper-level frontogenesis, which is most frequent in winter due to an increased baroclinicity (cf. Section 3.1.1), leads to tropopause foldings [99]. In these episodes the intrusions of stratospheric air into the troposphere destabilize this latter layer and may trigger convection [100–102]. The detection of a greater number of Ci and Cb pixels is therefore predicted to occur in the winter months, as shown in Figure 13b.

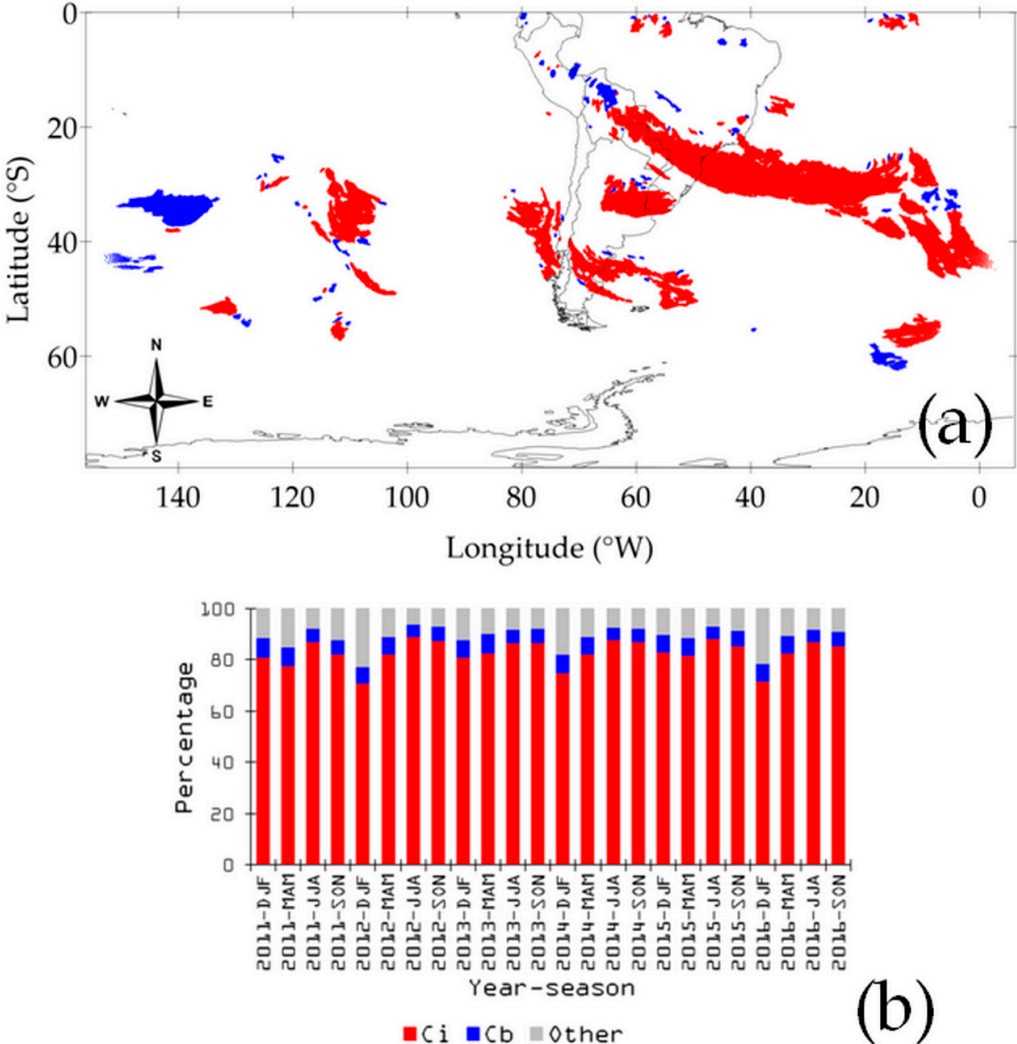

**Figure 13.** (**a**) Cluster 1's pixels identified as part of cirrus (Ci; red) and cumulonimbus (Cb; blue) clusters on 1 August 2011 and (**b**) mean seasonal percentages of Ci and Cb in cluster 1.

## 4. Discussion

A cluster classification of $T_b$ as observed by the GOES-13 in SA and adjacent oceans was carried out using a single IR band. As mentioned in Section 2, the complexity of the cloud dynamics together with the underlying surface properties and the absorption in the atmosphere implies a challenge to remotely sensed cloud measurements. Cloud detection and classification based on satellite-retrieved IR observations is helpful since it provides valuable information of the presence of clouds with a good coverage both spatially and temporally, yet there are a number of constraints for the accuracy in the retrieval of cloud properties for all type of clouds, but particularly for Cis. These clouds are often semitransparent in the IR spectrum and data derived from a single IR channel have limitations for the retrieval of both their temperature and their emissivity [103]. The size of the ice crystals found in these particular clouds range from 10 to 1000 μm [5]. Their effect in the calculation of $T_b$ can be twofold. The size range of the ice particles is comparable to the wavelength of the emission of the Earth-atmosphere system [5], therefore altering $T_b$ as calculated from the IR radiances and making their tops warmer and lower than they actually are. On the other hand, since ice particles can modify the upwelling radiation by scattering photons away from the satellite sensors' field of view [104] they may alter the actual $T_b$ values as well. The number of pixels represented by clusters 1 and 2 could also be greater than the ones informed as semitransparent Cis could be completely missed from the satellite retrievals [105] given that transmissive Cis may account for up to 42% [106]. Regarding this point,

another aspect worth considering is the presence of multilayered cloud decks [51], such as the ones in Figure 7a,f where Cis were clearly visible. Yet another important shortcoming when using a single IR channel is that different types of clouds that have the same $T_b$ value cannot be properly resolved [107]. Despite this, there is at least one precedent in the literature that used a single IR channel to detect clouds [107]. The authors then used the $T_b$ gradient from the water vapor channel to achieve the classification of high-level clouds. In conjunction with actual physical processes, a possible explanation for the highest variability in cluster 1 relies upon the fact that there is a greater density of ice particles in the clouds that populate this cluster. The lower-order clusters could also be affected by the same effect even when the density of ice particles decreases.

In spite of the aforementioned issues, which are inherent to the measurement process and can be mitigated but not eliminated, we managed to classify GOES IR imagery in four cloud regimes. In certain cases, if further partitions of the original clusters were intended the optimal number of new clusters as estimated using the VRC turned out to be 1. This suggested that no further actions could be taken from a k-means perspective and that alternative classification algorithms should be endeavored. Notwithstanding, the number of clusters being four is in agreement with the existing literature, in particular with the cloud regime classification carried out in another study region [76]. In principle, the four clusters can be associated to different types of clouds. For instance, cluster 4 seems to be associated to shallow clouds that span the lowest 2 km of the troposphere. On the other hand, cluster 1 tends to represent clouds whose tops populate the upper troposphere. Clusters 2 and 3 have their tops in the middle troposphere, and they could be resolved by using supplementary information such as cloud coverage, optical thicknesses, water vapor profiles, microphysical top phases or specific cloud type retrievals. However, the use of such tools does not guarantee a clearer resolution of the different types of the clouds involved owing to the fact that different clouds can share the same properties, or there can be multilayered clouds. The latter aspect applies to cluster 4 as well.

Another point that is worth mentioning is the use of vertical profiles to establish, through linear regressions, a relationship between temperature and height. Non-linear fits to the temperature profiles could also be conducted. Even though this alternative is expected to improve the outcome that relates the two variables, it is worth emphasizing that the results described in this paper are qualitatively in correspondence with the existing literature.

Regarding the partition of cluster 1 into Ci and Cb pixels the results is encouraging considering this was a first attempt. Yet, some types of high clouds (Ccs and Css), or even mixtures of all the high clouds that populate the analyzed cluster, were left unaccounted for due to the $A > 200$ km$^2$ criterion. The number of unaccounted objects peaked in summer. The seasonal differences could be partly diminished by conducting a seasonal analysis. However, the SDs of the D(P) values were low enough to expect a significant reduction in the seasonal differences. Separately, the methodology should be refined in order to include objects with hollow regions (a typical example of this is an entire patch of Ccs) by carrying out morphological operations to cluster 1. Other possible improvements include the use of alternative models to quantify object perimeters and the implementation of multifractal analyses.

The paper was aimed at classifying CC in SA and its adjacent oceans. The results regarding the cloud regimes are in agreement both with the different meteorological phenomena in the region and with the existing literature. Future research directions pursue the use of these results in analyses that include the identification of synoptic and sub-synoptic cloud systems and their relationships to multi-scale atmospheric processes.

## 5. Conclusions

An unsupervised clustering method based on the combination of the k-means and k-means++ algorithms was implemented on images of standardized $T_b$ anomalies derived from GOES-13 IR data for the period 1 December 2010 to 30 November 2016. The aim was to obtain a decomposition of each individual $T_b$ image into K clusters that capture the characteristics of different CC. The methodology

that was carried out allowed the estimation of an optimal number of four clusters (K = 4), each one associated to different cloud types.

Generally speaking, the lower the order of the cluster the higher the clouds it represents. Cluster 4 represented a cloud regime that spanned the lowest 2 km of the troposphere, and was the most dominant one representing, on average, 32% of the pixels. Likely, this cluster also captures cloud-like patterns that take place at surface levels or near it, i.e., fog [108], but this was not checked. Nevertheless, given that fogs can form up to 1000 m [108], cluster 4 may well represent this particular cloud type. Regarding clusters 2 and 3, there was an entanglement between them in the sense that both had their CTHs spanning the middle troposphere. Considering clusters 2 and 3 altogether they represented, on average, 62% of the pixels. The types of clouds that may populate these two clusters include altostratus, nimbostratus and stratocumulus. Finally, cluster 1 was the less dominant one (6% of the pixels on average), it represents clouds with the highest tops, and it showed the highest variabilities as well.

Even though cluster 1 was populated with clouds having different microphysical properties, most notably Cis and Cbs, the cluster methodology alone could not properly resolve these types of clouds. Concerning this, the inclusion of additional parameters into the analysis (e.g., optical thicknesses) will help in refining the classification method. In this paper, a fractal-related analysis permitted distinguishing between Cis and Cbs. The former cloud types were the most abundant across the year, accounting for 83% of cluster 1's pixels on average. Cbs represented 6% of cluster 1's pixels on average. The results were satisfactory but, since this technique relies upon the pixel size, the current analysis will benefit from the incorporation of higher resolution images.

**Supplementary Materials:** The following are available online at http://www.mdpi.com/2072-4292/12/18/2991/s1, Figure S1: Brightness temperature monthly mean values and standard deviations. These values were used to standardize the study variable pixel-by-pixel, Figure S2: Geopotential height at ten different levels from 850 hPa to 100 hPa on 1 August 2011. The wind field and the kinetic energy per unit mass at all of these levels are also shown. This figure was prepared using daily data from the NCEP/NCAR reanalysis [64].

**Author Contributions:** Conceptualization, A.E.Y., S.G.L., A.C. and P.O.C.; methodology, A.C. and J.P.M.; software, A.E.Y. and A.C.; validation, A.E.Y., S.G.L., A.C. and P.O.C.; formal analysis, A.E.Y., S.G.L. and A.C.; investigation, A.C.; resources, A.E.Y. and A.C.; data curation, A.C.; writing—original draft preparation, A.E.Y. and A.C.; writing—review and editing, A.E.Y. and S.G.L.; visualization, A.E.Y., S.G.L. and A.C.; supervision, A.E.Y. and S.G.L.; project administration, A.E.Y.; funding acquisition, A.E.Y., S.G.L. and P.O.C. All authors have read and agreed to the published version of the manuscript.

**Funding:** This research was partly funded by Argentina's Ministerio de Defensa, grant number PIDDEF 26/14, and Universidad Tecnológica Nacional SCyT and Facultad Regional grant numbers PID MSUTNBA0004570 and PID MSUTNBA0004787.

**Acknowledgments:** We gratefully acknowledge two anonymous reviewers for their comments and suggestions.

**Conflicts of Interest:** The authors declare no conflict of interest. The funders had no role in the design of the study; in the collection, analyses, or interpretation of data; in the writing of the manuscript, or in the decision to publish the results.

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
