# Peer review of "A Cluster Approach to Cloud Cover Classification over South America and Adjacent Oceans Using a k-means/k-means++ Unsupervised Algorithm on GOES IR Imagery"

_remotesensing, doi:10.3390/rs12182991_

Round 1
Reviewer 1 Report
Please, see the attached file.

Reviewer 2 Report
An excellent topic and well-designed research. The manuscript is very well written. The only observation I have is the last paragraph of section (2.2 The k-means/k-means++ clustering algorithm) which states “The regression coefficients were used to associate Tb values with CTHs. Negative CTHs that resulted from this regression were not included in the analysis.” I will appreciate if you could include the reason for not including negative values.
The other observation is the “conclusion section”. It is more like a summary of the research and may need to be rewritten drawing some definitive conclusion (may be about the technique used and its effectiveness).
The last suggestion is to include scale bar and north arrow in figures (4, 5, 6, 7, and 8) as it is usually a cartographic requirement. I believe figure 3 is not used in the manuscript and it may change figure number.
Thanks for a nice research work.
Round 2
Reviewer 1 Report
The manuscript is acceptable in the present shape since the authors addressed my requests.